# Male Tract Microbiota and Male Infertility

**DOI:** 10.3390/cells13151275

**Published:** 2024-07-29

**Authors:** Giuseppe Grande, Andrea Graziani, Luca De Toni, Andrea Garolla, Alberto Ferlin

**Affiliations:** Unit of Andrology and Reproductive Medicine, Department of Medicine, University of Padova, 35128 Padova, Italy; giuseppe.grande@aopd.veneto.it (G.G.); luca.detoni@unipd.it (L.D.T.); andrea.garolla@unipd.it (A.G.)

**Keywords:** microbiome, microbiota, male infertility, gut, prostate, prostatitis

## Abstract

In recent years, several studies have analyzed the composition of the male genital tract microbiota and its changes in infertility or in different situations associated with infertility. The aim of this narrative review is to obtain more insight on this topic; in particular, to describe actual evidence about changes in the semen microbiota in patients with infertility, male tract infections, or HPV infections. In semen, an increase in semen *Prevotella* spp. is associated with oligozoospermia and with obesity-associated asthenozoospermia; an increase in *Pseudomonas* is more frequently associated with asthenozoospermia and oligozoospermia; a reduction in *Lactobacilli* spp. (namely in *Lactobacillus crispatus*) may represent a marker of low semen quality. However, an increase in *Lactobacillus iners* is considered a risk factor for a reduced sperm concentration. In patients with prostatitis, there is a reduction in *Lactobacillus* spp. and an increase in *Streptococcus* spp., opening important perspectives about the role of probiotic treatments in these patients. Finally, an increase in *Fusobacteria* spp. was observed in patients with an HPV infection. In the conclusion, we underline the interactions between the seminal and vaginal microbiota, so that further studies should focus on the “couple genital microbiota”.

## 1. Introduction

The term “microbiota” has its roots in the early 1900s. Numerous microorganisms, such as bacteria, yeasts, and viruses, have been discovered to cohabitate in the human body’s many organ systems (gut, skin, lung, and oral cavity) [1]. It is said that we are born 100% human, but we die 90% microbial. Compared to the complete human genome, the human microbiota—also referred to as “the hidden organ”—contributes more than 150 times as much genetic information [2].

Even though the terms “microbiota” and “microbiome” are frequently used synonymously, they have some distinctions. The term “microbiota” refers to the living microbes that are present in a certain habitat; the term “microbiome” refers to the collection of genomes from all the microorganisms in the environment [3].

Next-generation sequencing technologies, which aim to target amplicon sequencing of the 16S-seq and enable the identification and quantification of human-resident microorganisms at unprecedented resolution, have revealed novel insights on the composition of the microbiota [4].

Site-specific variations exist in the microbiota’s composition. The microbiota in our gut is thought to be the most important for preserving our health [5]. The term “gut microbiota” refers to the community of microorganisms that live in the human gut. The gut microbiota is a dynamic system that varies during a person’s lifetime [6]. The five primary bacterial phyla that make up the human gut microbiota are: Firmicutes (60–80%), which includes the classes of *Bacteroidia*, *Negativicutes*, and *Clostridia*; Bacteroidetes (20–40%), which includes *Flavobacteria*, *Bacteroidia*, *Sphingobacteria*, and *Cytophagia*; Verrucomicrobia; Actinobacteria and, to a lesser extent, Proteobacteria; and one Archea phyla, the Euryarchaeota.

The preservation of immunological tolerance is significantly influenced by the gut microbiota. There is a dynamic and interdependent interaction between these microorganisms and the immune system. They generate immune signals by presenting a vast repertoire of antigenic determinants and microbial metabolites that influence the development, maturation, and maintenance of immunological function and homeostasis. Simultaneously, the pathogenic and commensal bacteria in a stable microbial ecology are regulated by both the innate and adaptive immune systems. As a result, the host immune system and the gut microbial community cooperate to support immunological homeostasis. Therefore, interference with such a balanced condition can have a significant impact on human health and results in a range of autoimmune and auto-inflammatory diseases [7]. Furthermore, gut dysbiosis is strongly associated with bowel permeability [8].

The gut microbiota plays a major role in modulating the immune system, influencing mental health, digestion and absorption in the gut, obesity, and metabolic disorders [9].

Furthermore, the gut microbiota and its products can modulate and affect multiple organs other than the gut, including the male reproductive system.

In the field of andrology, gut dysbiosis and the consequent increase in bowel permeability have been called into question as involving in some ways the pathogenesis of obesity-associated functional male hypogonadism, since an increase in lipopolysaccharide (LPS) has been demonstrated in these patients, correlated with testosterone levels [10]. Therefore, it has been proposed that the increase in endotoxin levels might represent a cause of hypogonadism in these patients, since it is known that endotoxin reduces testosterone production by the testis, both by direct inhibition of Leydig cell steroidogenic pathways and indirectly by reducing pituitary LH drive [10,11]. Endotoxin (LPS) has been, in fact, shown in numerous animal studies to have the potential to affect testicular function. It has been demonstrated that experimentally administering LPS to rats, sheep, cattle, and non-human primates reduces the frequency and amplitude of LH pulses by suppressing hypothalamic-pituitary function [11] Moreover, research on animals has verified that Leydig cells express the endotoxin-specific TLR4 [12], and that experimental administration of LPS directly suppresses the testosterone production of Leydig cells [13]. The reduction in Leydig cell expression of steroidogenic acute regulatory (StAR) protein activity [14], which is crucial for the first transfer of cholesterol into mitochondria for subsequent conversion to testosterone, is most likely the mechanism by which endotoxin directly inhibits the production of testosterone.

Furthermore, it has been shown that systemic endotoxemia and increased LPS can have a detrimental effect on the blood–testis barrier’s natural structure by rupturing cell junctions and inducing inflammatory damage. According to a previous study, exposure to LPS can cause orchitis and markedly reduce the expression of cell junction markers, including the gap junction α-1 protein, tight junction protein 1, and testicular intercellular adhesion molecule 1 [15]. Moreover, IL-6 produced by the gut microbiota can damage Sertoli cell tight junctions by interfering with the ERK-MAPK signaling pathway and altering the location and quantity of the constitutive proteins of the blood–testis barrier [16].

Previous studies have shown that the gut microbiota strongly contributes to the constitution of the microbiota in different districts, including the male urogenital tract [17]. In recent years, several studies analyzed the composition of male genital tract microbiota and its changes in infertility or in different situations associated with infertility (i.e., HPV infection, prostatitis). The aim of this narrative review is to obtaine more insight on this topic.

## 2. Materials and Methods

We conducted a literature review on PubMed, through December 2023, using the words “microbiota and male infertility”, “microbiome and male infertility”, “microbiome and prostatitis”, and “microbiome and semen” and filtering for original studies performed on humans.

## 3. Results

We collected 424 results. After excluding non-English-based results, case reports, other studies/reviews, studies without available manuscripts, and studies outside the topic of the present review (i.e., studies in patients with prostate or testis cancer), we obtained 27 results, which have been used for this narrative review.

## 4. Semen and Male Tract Microbiota

The conclusion of past research employing PCR- and culture-based microbial identification techniques was that infection, not commensal bacteria, is the cause of the bacteria’s abundance in semen [18]. The idea of a sterile male tract is being called into question by recent research employing 16S RNA-Seq techniques, which indicate that the semen may have its own microbial population.

Lundy SD et al. in 2021 investigated the connection between male reproduction and the genitourinary and gastrointestinal microbiomes. They discovered a rich microbiome shared between semen and voided urine, which is in line with previous studies. This study underlined, moreover, that, in line with other studies comparing the gut, vaginal, and urinary microbiomes in females, the microbiome found in the male gut differs significantly from that found in the urine and semen, which seem to be rather like one another (perhaps because of a major urethral contribution). The Proteobacteria members *Pseudomonas, Pseudoxanthomonas*, and *Acidovorax* were overrepresented in semen, indicating distinct contributions from upstream anatomic sites such the seminal vesicle, prostate, and/or testis [19].

Furthermore, significant changes in the microbiome have been reported among the different parts of the male tract, as reported in Figure 1. In detail, *Collinsella* (phylum Actinobacteria) and *Staphylococcus* (phylum Firmicutes) were both reduced in semen after a vasectomy, providing additional evidence for being part of a specific testicular/epididymal microbiome [19].

Further studies observed in testis a high presence of Firmicutes and Bacteroidetes. In detail, Molina et al. applied 16S ribosomal RNA gene sequencing and followed stringent decontamination protocols to testicular biopsies, to unravel the microbial composition in the testicle [20]. Ten bacterial genera, including *Blautia* (phylum Firmicutes), *Cellulosibacter* (Firmicutes), *Clostridium XIVa* (Firmicutes), *Clostridium XIVb* (Firmicutes), *Clostridium XVIII* (Firmicutes), *Collinsella* (Actinobacteria), *Prevotella* (Bacteroidetes), *Prolixibacter* (Bacteroidetes), *Robinsoniella* (Firmicutes), and *Wandonia* (Bacteroidetes) were all found to be specific to testicle sperm, thus confirming the prevalence of Firmicutes and Bacteroidetes in the testicular microbiota. Previous studies [21,22,23,24] have identified the detected bacteria *Blautia*, *Clostridium*, and *Prevotella* among the seminal samples. This indicates that the bacteria that are most abundant in the testicular sperm samples are also present in the semen, indicating that these bacteria may contribute to the composition of the seminal microbiome downstream. It is interesting to note that *Prevotella* was found in more than 90% of the testicular samples.

Furthermore, the fact that contamination accounted for 50–70% of all the bacterial reads found in testicular sperm samples is a significant finding, supporting the theory that assisted reproductive technology is not performed in sterile conditions [22] and it emphasizes the significance of accounting for potential contaminants when working with tissue with low microbial biomass [20].

On the other side, a higher presence of Cyanobacteria and Actinobacteria have been reported in prostate and seminal vesicle, as confirmed by previous studies in vasectomized men [19]. In detail, the abundance of *Corynebacterium* (phylum Actinobacteria) increased in the semen after a vasectomy, thus confirming its prevalence in accessory glands.

Actinobacteria are in detail the most expressed phylum in the prostate. When the seminal vesicles of wild-type mice were examined, a modest percentage of Cyanobacteria and a large quantity of Firmicutes were found in the microbiome [25]. Small quantities of Cyanobacteria have also been found in the seminal microbiome, making them a specific in vivo marker of the seminal vesicle microbiome [26].

Seminal plasma reflects therefore the contribution of all the “microbiotas” in the male tract, being normally composed in major part of Firmicutes (50–59%), Proteobacteria (19–25%), Actinobacteria (8–10%), and Bacteroidetes (5%) [27].

Although the exact function of bacteria in semen is unknown, their potential involvement in the regulation of immunological and inflammatory responses has been demonstrated in the gut [28], the human body’s most extensively researched commensal ecosystem. Like the gut mucosa, the male reproductive tract displays a full range of immune responses that are influenced by interactions between immune system components, sex hormones, and a distinct microbiome [29]. These responses are critical for both successful reproduction and defense against microbial invasion [26].

## 5. Male Infertility and Microbiota

The World Health Organization (WHO) defines infertility as being unable to conceive after 12 months or more of regular, unprotected sexual activity [30]. Between 20 and 50 percent of cases lack a known cause [31,32]. In this perspective, situations where no identifiable causes can be found, following a thorough and comprehensive diagnostic approach, it should be classified as “idiopathic infertility” [33,34]. This percentage of idiopathic infertility does demonstrate how little is known about the fundamental processes controlling spermatogenesis and sperm function [34,35]. The diagnosis of idiopathic infertility might be therefore used to define a disorder whose etiology has not been defined yet [35,36] or whose cause cannot yet be determined using the tools available for research. As criteria for male infertility are not uniform, the rate of idiopathic infertility varies on the diagnostic care and depth of analysis.

In this field, the analysis of microbiota might represent an interesting, novel tool for the identification of specific patterns of microbiota alterations, associated with testicular dysfunction and male infertility, especially in patients with idiopathic infertility.

When analyzing human semen samples, the *Prevotella* genus has been linked to low-quality semen, which suggests that some *Prevotella* species may be responsible for spermatogenesis impairment and male infertility. Since *Prevotella* in semen originates directly from the testis, it is of particular interest that its abundance in semen was inversely correlated with semen concentration [19]. Furthermore, *Prevotella* abundance is directly correlated with BMI [19], thus suggesting that it may represent an element involved in obesity-linked infertility. Further studies confirmed this evidence, demonstrating that an inverse correlation between sperm motility and BMI is present in obese subjects when associated with an increase in the ratio Prevotella:Bacteroidetes; in patients with a normal P:B ratio obesity was not associated with a reduction in sperm motility [37].

Semen from infertile patients express, moreover, a higher abundance in *Pseudomonas* spp. In detail, *Pseudomonas* abundance has been directly associated with total motile sperm count [37]. Furthermore, more recently it was demonstrated that men with an abnormal sperm concentration showed a higher abundance in semen of *Pseudomonas* and *Pseudomonas fluorescens* [38].

Lundy et al. used shotgun metagenomics to better explore the possible molecular underpinnings of the relationships between bacterial and sperm metabolomics, and the results showed that the S-adenosyl-L-methionine (SAM) cycle was significantly over-represented in the semen of infertile men with an over-representation in seminal *Pseudomonas* spp. [19]. SAM is a common metabolite that has functions in aminopropylation, oxidative stress, and methylation [39]. First, abnormalities in sperm DNA methylation are significantly linked to male infertility [40]. SAM is the primary cellular methyl donor and it is essential for maintaining the incredibly distinct methylation patterns observed in spermatozoa. Second, SAM is a strong regulator of oxidative stress via conversion to the antioxidant glutathione, which increases sperm motility in infertile men [41]. Third, through the aminopropylation pathway, which is essential for spermatogenesis and motility, SAM regulates the synthesis of polyamines like spermidine [42]. Even though it is still unclear whether any of these processes contribute to male infertility caused by the altered microbiota and no data have been provided until now to the best of our knowledge about the role of SAM production in *Pseudonomas* infections, these findings are intriguing, and more research is needed.

Not only *Pseudomonas*, but other seminal bacteria in the Protobacterium phylum have been associated with infertility. Chen et al. demonstrated that in the semen of patients with non-obstructive azoospermia (NOA), *Ruegeria* and *Donghicola* dominated the NOA microbiota, and that a reduction in *Lactobacilli* may represent a marker of worse semen quality [43].

*Lactobacillus* spp. revealed a positive association with male fertility and semen parameters. Weng et al. compared seminal microbiomes from fertile and infertile subjects, demonstrating a higher proportion of *Lactobacillus* and *Gardnerella* in the normal samples, while that of *Prevotella* was significantly higher in the low-quality samples. Unsupervised clustering analysis demonstrated that the seminal bacterial communities were clustered into three main groups: *Lactobacillus*, *Pseudomonas*, and *Prevotella*. Remarkably, most normal samples (80.6%) were clustered in the *Lactobacillus* predominant group. Among the different *Lactobacilli*, *Lactobacillus crispatus* was associated with normal sperm morphology. Furthermore, samples with a higher abundance of *Lactobacillus iners* were associated with a reduced sperm concentration, thus demonstrating that, in contrast with other *Lactobacilli*, *Lactobacillus iners* might be related to a worse fertility prognosis [38].

In conclusion, previous studies demonstrated that some microbiome patterns are more frequently associated with male infertility and seminal alterations. In detail, an increase in semen *Prevotella* spp. is associated with oligozoospermia and obesity-associated asthenozoospermia; an increase in *Pseudomonas* is more frequently associated with asthenozoospermia and oligozoospermia; a reduction in *Lactobacilli* may represent a marker of low semen quality. Furthermore, a reduction in *Lactobacillus crispatus* has been associated with low sperm morphology, while an increase in *Lactobacillus iners* is a risk factor for a reduction in sperm concentration.

## 6. Leukocytospermia, Bacterial Prostatitis, and Microbiota Changes

According to the National Institute of Health (NIH) consensus, prostatitis syndromes include infectious forms (acute, ABP, and chronic CBP), chronic prostatitis/chronic pelvic pain syndrome (CP/CPPS), and asymptomatic prostatitis (AIP) [44]. Less than 10% of men suffering from chronic prostatitis have a confirmed bacterial infection [45].

However, since CP/CPPS is linked to chronic pain, a lower quality of life, and male infertility, it represents an important problem in routine clinical practice [46]. The number of men with symptoms related to prostatitis ranges from 2.2 to 9.7% of men, with a mean prevalence of 8–8.2%, whilst other studies report a prevalence of 35–50% of prostatitis in men during their lifetime [46]. Furthermore, it is important to highlight that diagnosing CP/CPPS can be challenging because patients often have no symptoms or only few, aspecific symptoms, and semen samples or prostatic secretions are frequently clear of bacteria. As a consequence, a deeper understanding of the etiology of chronic prostatitis and the identification of markers (including microbiome) for this condition represent a pivotal challenge in clinical practice [47].

In 2017, Mandar R. et al. analyzed for the first time the changes in seminal microbiota in patients with CP (NIHIIIa and NIHIIIb), demonstrating a significant reduction of *Lactobacilli* in semen of patients with prostatitis. Proteobacteria comprised the higher proportions in prostatitis patients than healthy men, although this difference did not reach the significance level [48].

More recently, Yao et al. compared seminal microbiota from infertile subjects with a normal sperm count presenting or not presenting leukocytospermia. They observed that subjects with leukocytospermia have a different semen microbiota, with a reduction in *Lactobacilli* and an increase in the *Streptococcus* spp. [49].

Although some attempts have been performed in order to modulate the microbiota by the use of probiotics in patients with prostatitis [50], further studies are needed to understand if oral administration of probiotics may improve the semen microbiota and the clinical impact of microbiota modulation on seminal parameters in patients with prostatitis.

In addition, CP/CPPS might have pathogenic mechanisms similar to IBS. In particular, those mechanisms might be both primitive pathogenetic factors (such as intestinal dysbiosis) or secondary pathogenetic mechanisms (such as modified, commensal gut flora related to antibiotic therapy) [49].

## 7. Semen HPV Infection and Microbiota

A Human papillomavirus (HPV) infection is the most common sexually transmitted disease (STD) in males and females worldwide [51].

The discovery of HPV in semen has generated a lot of attention in recent years. HPV has been found in both exfoliated and sperm cells. When compared to HPV-negative patients, an HPV infection may have a deleterious impact on seminal parameters, such as a considerable decrease in progressive motility and sperm morphology and a large increase in the sperm DNA fragmentation index [52].

Furthermore, several in vitro experimental investigations revealed a potential role for HPV in the decline in implantation and pregnancy rates during assisted reproductive technology (ART) procedures [52].

Several data have been published on women, demonstrating the link between changes in cervical and cervicovaginal microbiota and an HPV infection. In fact, the disruptions in the cervical and cervicovaginal microbiota, characterized by a decrease in *Lactobacillus* and the overgrowth of other bacteria, might increase the risk of HPV persistence and the progression of cervical abnormalities. This alteration in the cervicovaginal microbiota has been linked to a higher likelihood of an HPV infection and cervical dysplasia [53].

Otherwise, only one study has attempted to correlate semen microbiota and HPV infection in males. In 2021, Tuominen et al. [54] analyzed the seminal microbiome in six HPV-positive fertile healthy men and 25 negative controls. Although the patient samples were reduced, the HPV-positive semen samples exhibited differences in the taxonomic composition of the bacterial microbiota at phylum level; *Fusobacterium* spp. was differentially expressed in HPV-infected patients. At family level, *Streptococcaceae*, *Peptostreptococcaceae*, *Veillonellaceae*, and *Moraxellaceae* were more abundant in the HPV-positive samples than in the HPV-negative samples. The HPV-positive samples presented a higher relative abundance of the genera *Streptococcus*, *Serratia*, *Dialister*, and *Peptostreptococcus* compared to the HPV-negative semen samples.

*Fusobacteria* are able to enter epithelial cells, colonize mucosae, and modify local immunity due to their virulence factors, which include the adhesion molecule FadA [55]. An altered expression of *Fusobacterium* spp. has been observed in the gut of HIV-infected and virally suppressed MSM with a concurrent HPV infection. Furthermore, the changes in the abundance of *Fusobacterium* spp. have been called into question in several districts as being associated with HPV infections, such as in the vagina of patients with HPV-related cervical cancer [53], in the oral microbiota in patients with HPV-related cancer [41], and in HPV-related anal cancer [56], thus confirming the hypothesis of the association between this specific microbic profile and an HPV infection, as observed also in semen.

*Fusobacterium* is linked, moreover, in colorectal cancer to a poor tumor differentiation, high disease stage, poor prognosis [57] and metastatic disease [58,59]. The mechanisms behind these associations have been reported to include: (1) TLR4-induced signaling pathway initiation that results in the secretion of inflammatory cytokines like TNF-alpha and IL-8 [60]; (2) NF-kB expression that promotes cell proliferation and inhibits apoptosis [61]; (3) inhibition of NK T-cells [62]; and (4) recruitment of myeloid-derived suppressor cells that suppress CD4 T-cells [63].

Therefore, although the well described study by Tuominen et al. [54] seems to indicate that also for semen HPV infection, a modulation of the microbiota, namely in *Fusobacterium* spp., might play a role in HPV infections. Further, wider studies are needed to confirm these preliminary observations.

## 8. Future Perspectives and Conclusions

Table 1 shows the different clinical conditions associated with specific changes in semen microbiota.

In conclusion, it is important to underline that both gut and male tract microbiota is strictly influenced by environmental factors such as diet, smoking, exposition to pollutants, etc. [64].

The gut microbiomes of people with widely disparate dietary habits differ, in fact, especially between vegetarians and people following a western diet, which is often high in sugar, refined carbohydrates, meats, saturated fats, and alcohol. An abundance of *Candida* spp. is linked to a diet high in carbohydrates. On the other hand, in healthy subjects, a diet high in protein is linked to a lower quantity of *Methanobrevibacter* and *Candida* species [65]. No studies have been yet published, to the best of our knowledge, about the correlation between dietary habits and seminal microbiota; although it has been demonstrated that healthy lifestyles, including a higher adherence to the Mediterranean diet [66], are associated with better semen parameters. Further studies are therefore needed to clarify the impact of diet on seminal microbiota.

Furthermore, previous studies reported that smoking is associated with specific changes in gut microbiota [67] and has a deleterious impact on sperm parameters and fertility [68]. This evidence opens interesting, new perspectives in order to study the correlation between smoking and alterations in semen microbiota.

Moreover, during sexual intercourse, partners share their genital tract microorganisms. This bidirectional transfer can affect the microbial composition of one or both partners [69]. Sexual activity can have a big impact on the vaginal microbiota, which is an “open organ”. Numerous studies have demonstrated that having multiple sexual partners, having frequent episodes of receptive oral sex, having receptive anal sex prior to vaginal sex, and having sex with an uncircumcised male partner can all alter the microbial communities in the vagina and increase the risk of bacterial vaginosis episodes [70]. Therefore, the male genital tract microbiota directly influences the partner’s one and vice versa [71].

Several studies demonstrated that the vaginal microbiota may have a pivotal role in modulating sperm cell function, since some bacteria (i.e., *Escherichia Coli*) may induce sperm agglutination and immobilization [72], while vaginal *Lactobacilli* may exert several positive effects on sperm functions [73]. Furthermore, it has been recently demonstrated that, in cervical mucus, short-chain fatty acids, which represent the metabolites of the local microbiota, modulate sperm migration through the sperm olfactory receptor 51E2 activity [74], thus confirming the pivotal role of the female tract microbiota in controlling sperm cell function.

In this perspective, some initial attempts have been performed to study both seminal and vaginal microbiota in patients performing ART for infertility [75]. Further studies will probably define the role—not only of the vaginal or seminal microbiota—but will also consider the role of a “couple genital microbiota” as a key factor, to be eventually modulated by probiotic treatment, in couples seeking fertility.

The use of probiotics and prebiotics in the area of couple infertility is a promising therapeutic perspective. A recent systematic review of randomized clinical trials about the administration of *Lactobacilli* in males with idiopathic infertility demonstrated that probiotic administration reduces semen reactive oxygen species (ROS) levels and induces a notable enhancement in sperm motility [76]. Possible approaches to modulate the microbiota include, moreover, the administration of prebiotics. The most characteristic prebiotics are oligosaccharides, galacto-oligosaccharides, and breast-milk oligosaccharides; they have the capacity in rats to raise *Bifidobacterium* and *Lactobacillus* [77]. In a triple-blind, randomized, placebo-controlled clinical trial performed in patients with idiopathic infertility, there was the simultaneous administration of probiotics, namely *Lactobacilli*, and fructooligosaccharides as a prebiotic. After 80 days, the authors detected an improvement in the sperm concentration, motility, normal morphology and a reduction in sperm lipid peroxidation [78]. Similar results have been observed in a study in which *Lactobacillus paracasei* was associated with arabinogalctan, oligo-fructosaccharides, and l-glutamine as prebiotics [79]. Although further studies are needed, these results open interesting perspectives in the use of probiotics and prebiotics for male factor infertility.

In conclusion, although there was a limited number of studies and taking together the existing evidence, our study highlighted the role of seminal microbiota as a major actor both in physiological and in pathological conditions, including male infertility, prostatitis, HPV infections, and we tried to describe the future directions of investigations in this intriguing topic in the future.

## Figures and Tables

**Figure 1 cells-13-01275-f001:**
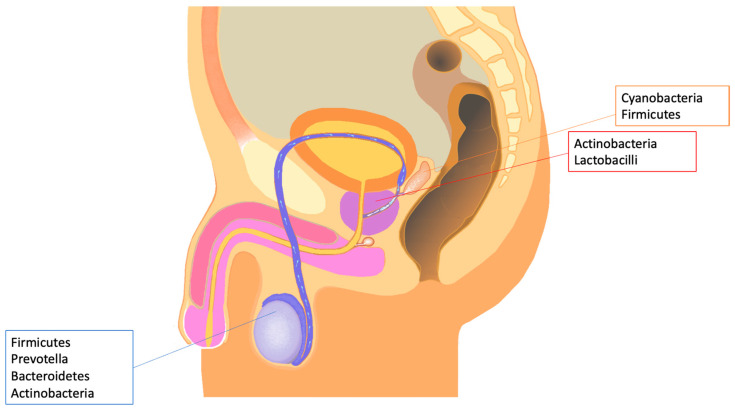
Major contributors to semen microbiota from testis, prostate, and seminal vesicles.

**Table 1 cells-13-01275-t001:** Association between changes in microbiota and seminal/clinical conditions.

Change in Seminal Microbiota	Seminal/Clinical Alterations	References
Increase in *Prevotella* spp.	Oligozoospermia	[19]
	Obesity-induced asthenozoospermia	[37]
Increase in *Pseudomonas* spp.	Oligozoospermia	[38]
	Asthenozoospermia	[37]
Increase in *Lactobacillus iners*	Oligozoospermia	[38]
Reduction in *Lactobacilli*	Reduction in all seminal parameters	[38,43]
	Prostatitis/Leukocytospermia	[48,49]
Increase in *Steptococcus* spp.	Leukocytospermia	[49]
Increase in *Fusobacteria* spp.	HPV infection	[54]

## Data Availability

No new data have been created.

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
