# Peer review of "Male Tract Microbiota and Male Infertility"

_cells, 2024, doi:10.3390/cells13151275_

Round 1

Reviewer 1 Report

Comments and Suggestions for Authors

The “narrative” review from Grande G et al aims to investigate the relationship among the composition of the seminal microbiota, male tract infections and male infertility. The authors have provided a detailed background on the topic and discussed the more relevant studies on the topic.

Nevertheless, some changes are strongly recommended.

1.        Introduction:  The section on gut microbiota is too long. It should be shortened and focused on the relevance of gut microbiota for immune homeostasis and health in general, on the relationships between the composition of the gut microbiota, obesity and hypogonadism. It should emphasize the concept that gut microbiota strongly contributes to the constitution of the microbiota in different districts, including the male urogenital tract and the relationship between gut and testes for the regulation of testicular function, the role of LPS in testicular damage, blood-testis barrier disruption etc, should be better defined.

2.        Methods:  even if it's a narrative review, information about the literary source selected for the study should be provided

3.        The relationship between vaginal microbiota, seminal microbiota and infertility should also be better defined as well as a short paragraph on the potential use of probiotics, prebiotics to treat male infertility should be included

4.        Finally, I believe that a table or integrating scheme on how selected microbial taxa affect sperm fertility potential will greatly help to summarize the information

Specific comments

Lane 27-29:

This indicates that each of us has 10x more microbe 27 cells in our body than human cells.

new work shows that the number of microbial cells is not much higher than that of human cells. Gene contribution from microbial world is more relevant than the number of microbial cells

Lane 34-36

the term "microbiome" refers to the collection of genomes from all the microorganisms in the environment._ _T_h_e_ _“m_i_c_r_o_b_i_o_m_e_” _comprises so the microbial population as well as metabolites, structural components, and environmental factors

The term microbiome indicates the large genetic contribution by the commensal microorganisms while “Metabolome” is the term used to indicate the metabolites (from all sources) in each microenvironment

45-46

The term "gut microbiota" refers to the community of microorganisms that live in the human gut, of which there are roughly 10 bacterial cells.

Do you mean 10 bacterial cells for each human cell?

lane 71-73: under healthy settings, Th1 and Th17 cells eliminate the translocation of a small number of bacterial products, suc as….

Under healthy settings Th1 and Th17 cells, activated by antigen presenting cells, contribute to maintain the integrity of epithelial barrier and to eliminate pathogens that eventually cross it.

SFBs are keystone species (of mouse gut microbiota) recognized as main actors of homeostatic immunity

Comments on the Quality of English Language

No comments

Author Response

1.        Introduction:  The section on gut microbiota is too long. It should be shortened and focused on the relevance of gut microbiota for immune homeostasis and health in general, on the relationships between the composition of the gut microbiota, obesity and hypogonadism. It should emphasize the concept that gut microbiota strongly contributes to the constitution of the microbiota in different districts, including the male urogenital tract and the relationship between gut and testes for the regulation of testicular function, the role of LPS in testicular damage, blood-testis barrier disruption etc, should be better defined.
We are grateful to the Reviewer for his/her observations. We modified the introduction according to your suggestions.
2.        Methods:  even if it's a narrative review, information about the literary source selected for the study should be provided
We modified the manuscript, reporting informations about the literary source research
3.        The relationship between vaginal microbiota, seminal microbiota and infertility should also be better defined as well as a short paragraph on the potential use of probiotics, prebiotics to treat male infertility should be included
We have discussed more in details the role of vaginal microbiota on sperm function and infertility and added a short paragraph about the potential use of probiotics and prebiotics in male infertility.
4.        Finally, I believe that a table or integrating scheme on how selected microbial taxa affect sperm fertility potential will greatly help to summarize the information
We added a resuming table about the correlation between different changes in microbial taxa and seminal/clinical conditions.

Reviewer 2 Report

Comments and Suggestions for Authors

This review summarizes the harmful and beneficial bacterial genera related to male infertility, and providing a new perspective for prevention or treatment in the field of reproduction. However, there are some issues with the logic, analysis, and discussion in the article, specifically as follows:

1. The main content of this article discusses the relationship between the male reproductive tract microbiota and infertility, but the term "Microbiota" in the title is too broad. For example, there is no information on effect of gut microbiota on reproduction in this paper. This paper mostly focus on male reproductive tract microbiota,It is recommended to narrow down the scope of the title to more specifically describe the relationship between the male reproductive tract microbiota and infertility.

2. Line 23: The focus of this article is on male infertility, not the microbiota. The introduction section starts by discussing the history of microbiota, which deviates from the main theme of the article. It is recommended to adjust the introduction to focus more on the relationship between male infertility and the microbiota, and reduce or eliminate the introduction of microbiota history that is not relevant to the main theme.

3. Line 31 and Line 45: In a review article, it is necessary to provide brief definitions of the terms "microbiota and microbiome" and "gut microbiota" to ensure that readers have a basic understanding of these fundamental concepts. However, it is even more important to cite and summarize existing research to demonstrate the applications and significance of these terms in actual research.

4. Line 44: This section primarily describes the relationship between gut microbiota and other diseases, which deviates from the main theme of the article regarding the relationship between microbiota and male infertility. It is recommended to delete or modify this section to make it more focused on the relationship between microbiota and male infertility.

5. Lines 93-103 contain an extensive description of semen composition, which is basic content. Please revise it.

6. Line 164: This section introduces the background of male infertility, which should be placed in the introduction to make the article structure more reasonable. Please revise it accordingly.

7. Line 198: This section does not discuss the relationship between SAM (specific microorganism or condition, depending on the context) and Pseudomonas spp. Please clarify this point.

8. Line 238: Please reduce the description of prostatitis and HPV diseases, and instead, add insights or a summary of the relationship between the microbiota and prostatitis and HPV.

Comments on the Quality of English Language

Authors should also conduct  Minor editing of the language in their manuscripts.

Author Response

This review summarizes the harmful and beneficial bacterial genera related to male infertility, and providing a new perspective for prevention or treatment in the field of reproduction. However, there are some issues with the logic, analysis, and discussion in the article, specifically as follows:
1.    The main content of this article discusses the relationship between the male reproductive tract microbiota and infertility, but the term "Microbiota" in the title is too broad. For example, there is no information on effect of gut microbiota on reproduction in this paper. This paper mostly focus on male reproductive tract microbiota,It is recommended to narrow down the scope of the title to more specifically describe the relationship between the male reproductive tract microbiota and infertility.
We are grateful to the Reviewer for his/her observations.
We modified the title according to your suggestion.
2.    Line 23: The focus of this article is on male infertility, not the microbiota. The introduction section starts by discussing the history of microbiota, which deviates from the main theme of the article. It is recommended to adjust the introduction to focus more on the relationship between male infertility and the microbiota, and reduce or eliminate the introduction of microbiota history that is not relevant to the main theme.
We modified the introduction, thus reducing the “historical” part about microbiota and focusing on the role of gut microbiota and male reproduction and the correlation between gut and male tract microbiota, as also suggested by the other Reviewer.
3.    Line 31 and Line 45: In a review article, it is necessary to provide brief definitions of the terms "microbiota and microbiome" and "gut microbiota" to ensure that readers have a basic understanding of these fundamental concepts. However, it is even more important to cite and summarize existing research to demonstrate the applications and significance of these terms in actual research.
We have added a reference for the definition of “gut microbiota”
4.    Line 44: This section primarily describes the relationship between gut microbiota and other diseases, which deviates from the main theme of the article regarding the relationship between microbiota and male infertility. It is recommended to delete or modify this section to make it more focused on the relationship between microbiota and male infertility. 
We modified the introduction, thus deleting the part describing the role of gut micriobiota in other diseases, while explaining more in details how gut microbiota strongly contributes to the constitution of the microbiota in different districts, including the male urogenital tract and the relationship between gut and testes for the regulation of testicular function, the role of LPS in testicular damage, blood-testis barrier disruption, as requested by Reviewer 1.
5.    Lines 93-103 contain an extensive description of semen composition, which is basic content. Please revise it.
We deleted this section.
6.    Line 164: This section introduces the background of male infertility, which should be placed in the introduction to make the article structure more reasonable. Please revise it accordingly.
We modified this section, deleting data exceeding from the aim of the current review.
7.    Line 198: This section does not discuss the relationship between SAM (specific microorganism or condition, depending on the context) and Pseudomonas spp. Please clarify this point.
As stated in the revised manuscript, “no data have been provided since now at the best of our knowledge about the role of SAM production in Pseudonomas infections” and further studies are needed to better clarify this aspect.
8.    Line 238: Please reduce the description of prostatitis and HPV diseases, and instead, add insights or a summary of the relationship between the microbiota and prostatitis and HPV.
We reduced the description of prostatitis and HPV disease according to your suggestion. Furthermore, we added to conclusion a table reporting the major reports among the association between alterations in semen microbiota and different clinical conditions, including prostatitis and HPV infections.

Comments on the Quality of English Language Authors should also conduct  Minor editing of the language in their manuscripts.
We thank the Reviewer. A native English speaker revised and approved the final text. 

Reviewer 3 Report

Comments and Suggestions for Authors

Similar reviews have been published multiple times recently (Tomaiuolo R, et al. High Throughput. 2020,9(2):10; Venneri MA, et al. J Endocrinol Invest. 2022,45(6):1151; Magill RG, et al. Front Reprod Health. 2023,5:1166201), so this manuscript is lack of novelty.

Comments on the Quality of English Language

Moderate editing of English language required.

Author Response

Similar reviews have been published multiple times recently (Tomaiuolo R, et al. High Throughput. 2020,9(2):10; Venneri MA, et al. J Endocrinol Invest. 2022,45(6):1151; Magill RG, et al. Front Reprod Health. 2023,5:1166201), so this manuscript is lack of novelty.

We do not agree with the opinion of the Reviewer. First of all, it is an invited review about this topic. Secondly, it has several aspects of novelty, as compared with the cited reviews: we analyzed the role of changes in microbiota from a clinical perspective, as markers of different clinical conditions, including infertility and altered seminal parameters, but also – data missing in the 3 cited reviews– in patients with prostatitis/leukocytospermia and HPV infection. This perspective seems to us that offers an interest aspect of novelty to a widely discussed topic.

Round 2

Reviewer 1 Report

Comments and Suggestions for Authors

The revised form of the review from Grande G et al is much improved compared to the first one.

However, minor revisions need to be made. Below you can find some suggestions

Lane 41-43

Next-generation sequencing technologies, which aim to target amplicon sequencing of the 16S-seq and enable the identification and quantification of human-resident microorganisms at unprecedented resolution, have revealed novel insight on the composition of microbiota[4].

Lane 46:  cancel: “of which 46 there are roughly 10 bacterial cells”

Lane 276 Among the different Lactobacilli, Lactobacillus crispatus was associated with normal sperm morphology. Furthermore, samples with a higher abundance of Lactobacillus  iners were associated with reduced sperm concentration, thus demonstrating that, in contrast with other Lactobacilli, Lactobacillus iners might  be related with worse fertility prognosis

Lane 358

the disruptions in the CM and CVM.  These abbreviations were never used in the text. Better to use cervical and cervicovaginal microbiota.

If you use abbreviations, insert a paragraph with abbreviations frequently used. In the text write the complete name (i.e. cervicovaginal microbiota) followed by abbreviation (CVM) when use it for the first time

Table 1: references shoud be included in an additional column

The names of microbial taxa must necessarily be written in italics

Comments on the Quality of English Language

/

Author Response

We are grateful to the Reviewer for his/her observations. We have modified the manuscript according to his/her suggestions.

Reviewer 2 Report

Comments and Suggestions for Authors

This manuscript describes changes in semen microbiota in patients with infertility, male tract infections . All the issues have bee solved by the author.

Author Response

Thank you.

Reviewer 3 Report

Comments and Suggestions for Authors

The author has made significant revisions to this article, and I think it can be accepted in present form.

Author Response

Thank you.